# Magnesium ions mediate ligand binding and conformational transition of the SAM/SAH riboswitch

Guodong Hu [1,2] & Huan-Xiang Zhou [2,3✉]

The SAM/SAH riboswitch binds S-adenosylmethionine (SAM) and S-adenosylhomocysteine (SAH) with similar affinities. $Mg^{2+}$ is generally known to stabilize RNA structures by neutralizing phosphates, but how it contributes to ligand binding and conformational transition is understudied. Here, extensive molecular dynamics simulations (totaling 120 μs) predicted over 10 inner-shell $Mg^{2+}$ ions in the SAM/SAH riboswitch. Six of them line the two sides of a groove to widen it and thereby pre-organize the riboswitch for ligand entry. They also form outer-shell coordination with the ligands and stabilize an RNA-ligand hydrogen bond, which effectively diminishes the selectivity between SAM and SAH. One $Mg^{2+}$ ion unique to the apo form maintains the Shine–Dalgarno sequence in an autonomous mode and thereby facilitates its release for ribosome binding. $Mg^{2+}$ thus plays vital roles in SAM/SAH riboswitch function.

[1] Shandong Key Laboratory of Biophysics, Dezhou University, Dezhou 253023, China. [2] Department of Chemistry, University of Illinois Chicago, Chicago, IL 60607, USA. [3] Department of Physics, University of Illinois Chicago, Chicago, IL 60607, USA. ✉email: hzhou43@uic.edu

Although RNAs are best known for transferring genetic information from DNA to proteins, some RNAs such as riboswitches perform signaling and catalytic functions much like proteins[1]. Riboswitches, mostly found in bacteria, are located in the 5′-untranslated regions of mRNAs and typically consist of two domains: aptamer and expression platform[2,3]. The aptamer domain binds ligands such as metabolites and triggers the expression platform to turn on or off gene expression[4,5]. S-adenosylmethionine (SAM; Fig. 1a) is the major methyl donor for the methylation of nucleic acids and proteins and is thus an essential metabolite[6–9]. It consists of an aminocarboxypropyl, a positively charged sulfonium center substituted by methyl, and a 5′-deoxyadenosyl. After donating its methyl, SAM is converted to the neutral compound S-adenosylhomocysteine (SAH; Fig. 1a). Based on structure, sequence, and evolutionary relatedness, SAM-sensing riboswitches have been divided into several families[10,11], including SAM-I[12,13], SAM-II[14], SAM-III[15], SAM-IV[16], and SAM-I/IV[11], SAM-V[17], SAM-VI[18,19], and SAM/SAH[11,20]. SAM/SAH riboswitches are distinct by their similar affinities for binding SAM and SAH, and by their smaller size and lower complexity of the ligand-binding pocket[11,20].

An NMR structure of the env9b SAM/SAH riboswitch bound with SAH was recently determined by Weickhmann et al. (Protein Data Bank entry 6HAG)[20]. The structure features an H-type pseudoknot (Fig. 1b, c), which consists of two stems: A-form stem S1 formed by pairing the G1 to C5 bases with the C24 to G20 bases, and pseudoknotted stem S2 with five canonical pairs between G10, C11, U12, C14, and C15 in the 5′ strand and C42, G41, A40, G39, and G38 in the 3′ strand. Two short loops, L1 (A6 to G9) and L2 (U16 to C19), connect the S1 and S2 stems; a third long, flexible loop, L3 (A25 to U37), connects the 3′ end of S1 to the 5′ end of S2. A sandwich-shaped ligand-binding pocket is formed between the C8:G17 base pair and the C15:G38 base pair (Fig. 1c). The nucleobases lining the ligand-binding pocket mostly interact with the 5′-deoxyadenosyl group of the ligands. This

group is stacked between G38 on the ceiling and C8:G17 on the floor. The sugar portion of the 5′-deoxyadenosyl group forms both CH/π interactions with the base of G9 on the ceiling and hydrogen bonds with the phosphate-sugar backbone of G9. The base portion of the 5′-deoxyadenosyl group forms a reversed Hoogsteen base pair with the U16 base. Recent crystal structures of the SK209-2-6 SAM/SAH riboswitch (lacking the highly flexible nucleotides 26–34 of L3) bound with SAM or SAH are very similar to the NMR structure of the SAH-bound env9b SAM/SAH riboswitch[21].

At the 3′ end of the riboswitch, the Shine–Dalgarno (SD) sequence (G38-G39-A40-G41) can be freed to bind with the ribosome and initiate translation when no ligand is present[11,20,21]. As just noted, G38 base-stacks with the ligand. G39 forms a base triple with two other nucleobases, G9 and C14 (Fig. 1c). As G9 interacts with the ligand, it serves as a bridge between the ligand and G39. Through these direct and indirect interactions, the ligand sequesters the SD sequence and maintains the riboswitch in the translational off state. In the apo form of the env9b riboswitch, broad peaks in the imino proton NMR spectrum suggested that the riboswitch is only partially structured and conformationally heterogeneous[20]. Likewise, for the SK209-2-6 riboswitch, in-line probe indicated that nucleotides that form the pseudoknotted stem become less accessible upon SAM binding[11]; single-molecule FRET revealed that the apo form is in dynamic exchange between partially and fully folded conformations[21].

Because electrostatic repulsion of the phosphate backbone of RNAs would lead to unstable tertiary structures, cations are essential to provide charge neutralization[22]. Due to its small size and double charge, $Mg^{2+}$ is special for RNA structural stability, e.g., by forming bidentate coordination with two adjacent phosphates[23]. Moreover, $Mg^{2+}$ has been implicated in many other roles, including promoting folding or conformational transition[24–28] or rescuing misfolding[29], mediating ligand binding[20,30–36], and participating in catalysis[37]. In particular, although imino proton NMR spectra demonstrated that the env9b SAM/SAH riboswitch is capable of binding SAH in the absence of $Mg^{2+}$, isothermal titration calorimetry measurements showed that $Mg^{2+}$ significantly increases the binding affinities of the riboswitch for both SAH and SAM[20]. However, identifying $Mg^{2+}$ ions in RNA structures by experimental approaches is very challenging. Although the diamagnetic $Mg^{2+}$ in theory can lead to changes in NMR spectra of RNA, the effects may be small and challenging to resolve[38]. For example, the NMR structure 6HAG, though determined on samples in 2 mM $Mg(OAc)_2$, has no information on $Mg^{2+}$. For X-ray diffraction, because $Mg^{2+}$, $Na^+$, and water molecule all have the same number of electrons, distinguishing them is difficult and requires high resolution[39]. The total number of $Mg^{2+}$ ions in non-ribosome RNA crystal structures at worse than 2.1-Å resolution is very low. The 1.7-Å structure 6YL5 of the SK209-2-6 SAM/SAH riboswitch, crystallized in a buffer containing 10 mM $MgSO_4$ and 50 mM sodium cacodylate, resolved no $Mg^{2+}$ but a few $Na^+$ ions forming inner-shell coordination with the RNA[21].

A variety of computational approaches have been developed to predict metal ion binding sites in RNA structures or applied to elucidate the roles of $Mg^{2+}$. For example, MetalionRNA (http://metalionrna.genesilico.pl/) uses a statistical potential to predict metal ions inside RNA structures[40]; MCTBI (http://rna.physics.missouri.edu/MCTBI) predicts tightly bound ions by Monte Carlo sampling[41,42]; and MgNET is a convolutional neural network model trained on a set of crystal structures containing RNA and $Mg^{2+}$ ions[43]. Molecular dynamics (MD) simulations have been used to predict or characterize $Mg^{2+}$ binding sites in RNA[44–51]. When $Mg^{2+}$ ions are initially placed in the solvent[44,46,50,51], it can be difficult for them to form inner-shell

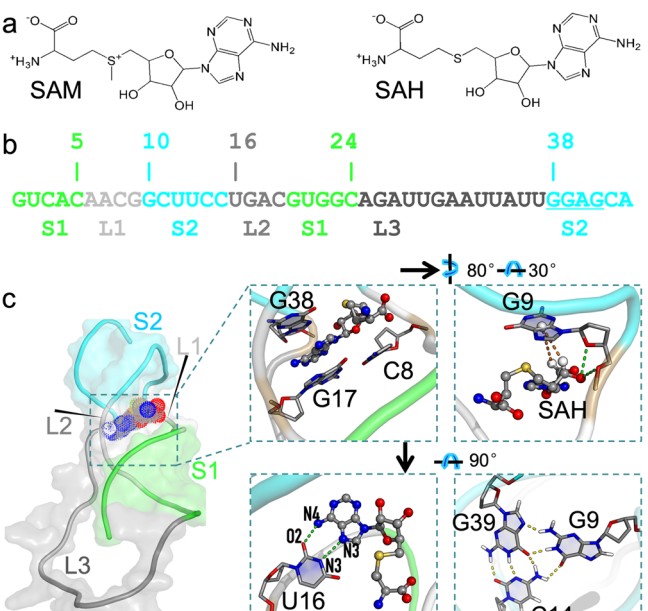

**Fig. 1 The structures of the SAM/SAH riboswitch and its cognate ligands. a** Chemical structures of SAM (left) and SAH (right). **b** Sequence and secondary structure of the riboswitch. The Shine–Dalgarno sequence is underlined. **c** Structure of the SAH-riboswitch complex (model 5 in 6HAG). Stacking and in-plane hydrogen bonding are highlighted in three zoomed views. The fourth zoomed view shows the base triple formed by G9, C14, and G39.

coordination with RNA during typical MD simulation times, due to the high barrier for dehydrating $Mg^{2+}$ [46]. Different approaches have been developed to enable the sampling of inner-shell sites by $Mg^{2+}$, including bias-exchange metadynamics[45], the use of an effective potential[27], and grand canonical Monte Carlo[28]. By initially placing $Mg^{2+}$ ions at sites predicted by a structure-based method such as MCTBI, we have demonstrated success in achieving $Mg^{2+}$-RNA inner-shell coordination in MD simulations[48]. MD simulations have also shown that $Mg^{2+}$ can promote the conformational transition of an RNA[26], quench conformational fluctuations[49,50], and stabilize ligand binding[48].

Here we carried out 120 μs of MD simulations on the env9b SAM/SAH riboswitch (Supplementary Table 1) to uncover how $Mg^{2+}$ mediates ligand binding and conformational transition. Eleven $Mg^{2+}$ ions stably form inner-shell coordination with backbone phosphates of the riboswitch, whether it is in the apo form or bound with SAM or SAH; nine of these sites are common among the three forms. Six of the common sites line either side of a groove that provides the entryway for the ligand. In the apo form, $Mg^{2+}$ ions at these sites widen the groove and thus pre-organize the riboswitch for ligand binding. Once the ligand is bound, three of these $Mg^{2+}$ ions can alternately form outer-shell coordination with the carboxy moiety of the ligands and also stabilize an additional U16-ligand hydrogen bond. These interactions have the effect of diminishing the selectivity between SAM and SAH. A unique inner-shell $Mg^{2+}$ ion in the apo form maintains a curved shape for the G38-G39-A40 backbone, thereby loosening their base-pairing with C15-C14-U12 and facilitating the release of the SD sequence for ribosome binding.

## Results

### Nucleotide-ligand interaction energies and ligand exposure clarify why the riboswitch is not selective between SAM and SAH.

Our MD simulations used the SAH-bound NMR structure 6HAG as the initial structure. We replaced SAH with SAM to generate the SAM-bound form and removed SAH to generate the apo form. Following our previous study[48], we tested three protocols for the initial placement of $Mg^{2+}$ ions in our RNA systems (Supplementary Table 1). Using MCTBI[42], we found 25 putative tight-binding sites and placed $Mg^{2+}$ ions at all these sites in the initial structure for MD simulations. Using the Leap module in AMBER18[52], we added 21 $Mg^{2+}$ ions around the RNA [Leap(21)], enough to neutralize the charges on the RNA. In the above two protocols, $Na^+$ ions (as part of the 0.15 M NaCl salt) were placed into the solvent. The third protocol also relied on Leap but we added $Na^+$ ions around the RNA and placed 41 $Mg^{2+}$ ions into the solvent [Leap(41)]. We also studied the case where the RNA and ligand molecules were only neutralized by $Na^+$ and no other ions were present (modeling the $Mg^{2+}$-free condition).

With the Leap(41) protocol, we ran four replicate 1-μs simulations of the liganded forms starting from each of the 10 models in 6HAG, and calculated the interaction energies of the 43 nucleotides of the riboswitch with the ligand by the MM/GBSA method[53] (Fig. 2a). Consistent with the binding pocket characterized by the NMR and crystal structures (Fig. 1c), only five nucleotides, C8, G9, U16, G17, and G38, make major contributions to the ligand binding energy. The interaction energies of each nucleotide with the two ligands are very close; however, they do favor SAM binding slightly but systematically, except for U16. The general favorability of SAM can be attributed to the long-range electrostatic attraction between the positive charge on its sulfur center and the RNA phosphates.

The MD simulations started from the different NMR models produced very similar interaction energies. We calculated

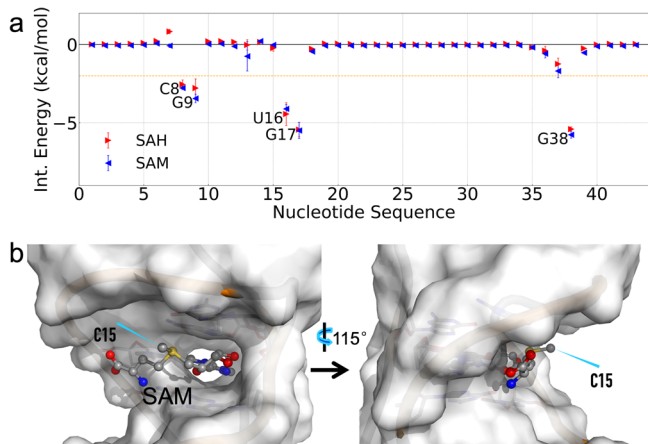

**Fig. 2 Lack of selectivity between SAM and SAH and a major reason. a** Interaction energies of individual nucleotides with the ligands. For each nucleotide, triangle and error bar represent the mean and standard deviation, respectively, calculated among results from 10 starting models. A horizontal line at −2.0 kcal/mol separates out the five pocket-lining nucleotides. **b** The solvent exposure of the SAM aminocarboxypropyl group, especially the methyl (C15), illustrated by a snapshot from the MD simulations under saturating $Mg^{2+}$. The backbone of the groove-lining nucleotides 5–8 and 12–16 is in orange.

Pearson's correlation coefficient ($r$) between the interaction energies from any two starting models (Supplementary Fig. 1). For both ligands, all the pairwise correlation coefficients are close to 1, with a minimum of 0.93 between any two models. The 10 models mostly differ in the conformations of the L3 loop (A25 to U37). As can be seen in Fig. 2a, except for the last two nucleotides (U36 and U37), L3 practically contributes no interaction energy with either ligand. Given the null effect of the starting model on nucleotide-ligand interaction energies, we limited to a single model in subsequent simulations. We chose model 5 because it has the smallest root-mean-square-deviation from the average structure of the 10 models.

Compared to the nucleotide-ligand interaction energies obtained from the simulations with Leap(41) protocol (Fig. 2a), the results with the MCTBI protocol are overall similar but the differences between SAH and SAM are greater for all the five major nucleotides: C8, G9, U16, G17, and G38 (Supplementary Fig. 2a). Moreover, G17 and G38 now join U16 in favoring SAH, thereby countering the general favorability of SAM. These changes can mostly be attributed to the effects of $Mg^{2+}$-phosphate coordination, since the nucleotide-ligand interaction energies from the Leap(41) protocol closely match those from the $Mg^{2+}$-free simulations (Fig. 2a and Supplementary Fig. 2b). The extent of $Mg^{2+}$-phosphate coordination with the Leap(21) protocol is intermediate between those of the MCTBI and Leap(41) protocols (see next subsection), and correspondingly the nucleotide-ligand interaction energy results are also intermediate (Supplementary Fig. 2c), in terms of the magnitudes of the differences between SAH and SAM for the major nucleotides and in terms of the number of major nucleotides (U16 and G17, but not G38) that run counter to the general favorability of SAM. In the simulations with the MCTBI protocol, G38 tends to move away from the ligand in the SAM-bound form, thereby explaining why this nucleotide favors SAH binding. The reason for the counteraction of U16 and G17 will be presented below. In short, under different extents of $Mg^{2+}$-phosphate coordination, nucleotide-ligand interaction energies have relatively small differences between SAH and SAM and even these small

differences are counteractive. These interaction energy results rationalize why the riboswitch has only a modestly higher binding affinity for SAM than for SAH ($K_D = 1.5$ μM and 3.7 μM for the two ligands)[20].

Below we focus on the results from the simulations started with the MCTBI protocol, to model the condition where the RNA is saturated with $Mg^{2+}$ and to draw the most contrast with the $Mg^{2+}$-free condition. While the 5′-deoxyadenosyl group of the ligand is buried in the binding pocket, the aminocarboxypropyl group and the sulfur center are exposed to a groove defined by nucleotides 5–8 and 12–16 (Fig. 2b). In the MD simulations of the SAH-bound form, the ligand has a "U" shape (Supplementary Fig. 3a). In the simulations of the SAM-bound form, the ligand switches between two conformations, one is similar to the U shape of SAH (Supplementary Fig. 3b) and the other has an "L" shape (Supplementary Figure 3c). In the U-shape conformation, the methyl on the sulfur center of SAM is the most exposed to the solvent (Fig. 2b). In the L-shape conformation, the aminocarboxypropyl group also extends into the solvent. Therefore, although the methyl plays a central role in distinguishing from SAH in SAM-sensing riboswitches, it loses this ability in the SAM/SAH riboswitch by projecting into the solvent.

The high flexibility of Loop L3, already evident from its different conformations in the NMR models, was directly assessed by $^1H$-$^{13}C$ heteronuclear Overhauser effects (hetNOE) in the SAH-bound form[20] (Supplementary Fig. 4a). From the MD simulations, we calculated the root-mean-square-fluctuations (RMSFs) of the corresponding atoms, i.e., aliphatic H1′/C1′ or aromatic H6/C6 (for C and U nucleotides) and H8/C8 (for A and G nucleotides) (Supplementary Fig. 4b). The results agree well with the hetNOE data. That is, L3 and terminal nucleotides show elevated flexibilities, and both U13 and U37 show higher flexibilities than their immediate neighbors. The flexibility of U13 can be explained by the fact that, although it is a part of stem S2, it only participates in a base triple with the U12:A40 base pair. As for U37, our MD simulations reveal that this base samples two alternative poses, as further described below. Relative to the liganded forms, the apo form exhibits higher RMSFs at both U13 and U37, capturing to some extent its reported conformational heterogeneity[11,20,21].

**SAM/SAH riboswitch can harbor over 10 inner-shell $Mg^{2+}$ ions.** With the MCTBI protocol, we found 11 $Mg^{2+}$ ions that are stably bound to inner-shell sites in each of the three forms (apo, SAH-bound, and SAM-bound) of the riboswitch throughout the simulations. With the Leap(21) protocol, the numbers of inner-shell $Mg^{2+}$ ions reduced to 6, 9, and 8, respectively (Supplementary Fig. 5); about one third of these ions are at the same sites as found in the simulations started with the MCTBI protocol. None of the $Mg^{2+}$ ions was initially at the inner-shell sites. Instead, they move to these sites during the energy minimization and the heating stage of the MD simulations, as demonstrated in Supplementary Fig. 5 and Supplementary Movie 1. With the Leap(41) protocol, not a single inner-shell $Mg^{2+}$ ion was found, because in this case $Na^+$ ions were initially added near the RNA and hence took up positions around phosphates. This pre-occupation by $Na^+$ ions prevents $Mg^{2+}$ ions, initially placed into the solvent, from moving into inner-shell sites, as observed in many other MD simulation studies[44–46,50,51]. In contrast to inner-shell $Mg^{2+}$ ions, $Na^+$ ions around phosphates are very mobile and do not stay in specific sites.

To identify the RNA atoms that form inner or outer-shell coordination with $Mg^{2+}$ ions, we calculated the radial distribution functions (RDFs) of $Mg^{2+}$ around oxygen and nitrogen atoms on the backbone and bases. Inner and outer-shell

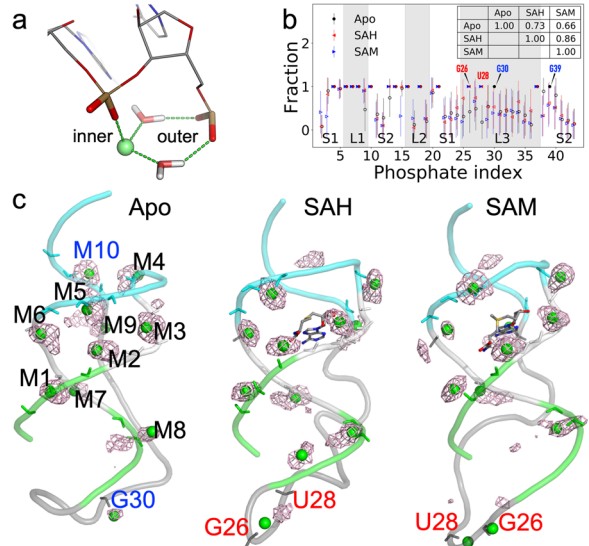

**Fig. 3 Inner-shell $Mg^{2+}$ ions in MD simulations of the apo and two liganded forms. a** Illustration of a $Mg^{2+}$ ion forming both inner-shell coordination with one phosphate and outer-shell coordination with an adjacent phosphate. Hydrogen bonds are indicated by dashed lines. **b** The fraction of frames where a phosphate forms inner or outer-shell coordination. Inner-shell coordination, once formed, is stable in the MD simulations (fraction = 1; solid symbols). For outer-shell fractions, open symbols and error bars represent the means and standard deviations, respectively, calculated among results from four replicate simulations. Inset table: correlation coefficients between any two forms of the riboswitch. **c** Densities of $Mg^{2+}$ ions, displayed as mesh and superimposed on a representative snapshot from the MD simulations. The inner-shell $Mg^{2+}$ ions are shown as green spheres, and the coordinating phosphates are shown as sticks.

coordination can be identified from RDF peaks at 2.0 Å and 4.3 Å, respectively (Fig. 3a and Supplementary Fig. 6). Inner-shell coordination is nearly exclusively formed with OP1 and OP2, with OP2 favored over OP1 by 1.3 to 3.0-fold (Supplementary Fig. 6). The preference for OP2 over OP1 arises from the fact that OP2 typically points toward the nuclease base whereas OP1 toward the solvent. Outer-shell coordination is most frequently formed with OP1, OP2, O3′, and O5′ on the backbone, and less frequently with bases. Of the latter, the N7 atom of the A base, N7 and O6 of G, and O4 of U are the most frequent. These inner and outer-shell statistics obtained from our MD simulations of the riboswitch in the apo and liganded forms are in remarkable agreement with $Mg^{2+}$ coordination frequencies tabulated from crystal structures in ref. [39].

In Fig. 3b, we present the fraction of MD frames where each nucleotide forms inner or outer-shell coordination. Most of the inner-shell sites are found in the first 21 nucleotides, containing stem S1, loops L1 and L2, and the 5′ strand of stem S2, and are largely conserved among the apo and two liganded forms. On the RNA surface, these nucleotides have the most negative electrostatic potential (Supplementary Fig. 7). The coordination patterns in the remaining 22 nucleotides show differences between the apo form and the two liganded forms. Overall, the correlations of the fraction values are strong between the two liganded forms ($r = 0.86$) but moderately reduced between either liganded form and apo (inset table). In Fig. 3c, we display the densities of $Mg^{2+}$ ions around the RNA, superimposed on a representative snapshot of inner-shell $Mg^{2+}$ ions.

For each inner-shell $Mg^{2+}$ ion, we identified its coordinating phosphates by calculating the distributions of their distances

(Supplementary Figs. 8 and 9). We name $Mg^{2+}$ ions that form inner-shell coordination outside the flexible portion of loop L3 as M1 to M10 (Fig. 3c). A few $Mg^{2+}$ ions form inner-shell coordination simultaneously with two adjacent phosphates, in a bidentate configuration[23], such as M3 with A7 and C8, or only with a single phosphate, such as M7 with A18. However, most inner-shell $Mg^{2+}$ ions also form outer-shell coordination with an adjacent phosphate. Outer-shell coordination occurs most frequently with both OP1 and OP2 of the adjacent phosphate via two bridging water molecules, as illustrated in Fig. 3a, but can also occur with only OP1 or OP2, via either one or two bridging water molecules. M1-M8 coordinate with the first 21 nucleotides and are largely conserved among the simulations of the three forms of the riboswitch (Supplementary Fig. 8). The flexible portion of L3 harbors one inner-shell $Mg^{2+}$ ion, coordinating with G30, in the apo form, but two other inner-shell $Mg^{2+}$ ions, coordinating either G26 or U28, in the liganded forms (Supplementary Fig. 9). These $Mg^{2+}$ ions move with the highly flexible L3, and, therefore, their densities are smeared out in space. U37 (nominally on L3) harbors the last conserved inner-shell $Mg^{2+}$ ion M9, and the 3′ strand of stem S2 harbors one last inner-shell $Mg^{2+}$ ion, M10, only in the apo form. Below we present further details of these $Mg^{2+}$ ions and their structural and functional consequences.

**Inner-shell $Mg^{2+}$ ions widen a groove and pre-organize the riboswitch for ligand entry.** As presented above, the 5′-deoxyadenosyl group of the ligand is buried in the binding pocket, with the groove defined by nucleotides 5–8 and 12–16 providing the entryway. We used the distance, $d_{P6-P14}$, between the phosphorus atoms of A6 and C14, to measure the groove width (Fig. 4a). The groove width increases upon ligand binding, both when $Mg^{2+}$-free or with saturating $Mg^{2+}$ (Fig. 4b). Interestingly, in the apo form, the mean groove width is increased, from 10.9 Å without $Mg^{2+}$ to 12.7 Å under saturating $Mg^{2+}$. The resulting groove width is comparable to those in the liganded forms without $Mg^{2+}$ (mean $d_{P6-P14}$ at 12.9 Å and 12.1 Å, respectively, for the SAH and SAM-bound forms). $Mg^{2+}$ saturation also results in groove widening in the liganded forms.

It thus appears that inner-shell $Mg^{2+}$ ions widen the ligand-entry groove and thereby pre-organize the riboswitch for ligand entry. Six conserved inner-shell $Mg^{2+}$ ions, M1–M6, line the two sides of this groove (Fig. 4c). The minimum distance between M2 and M3 (coordinating with C5–C8) on one side and M5 and M6 (coordinating with U13–U16) on the opposite side mirrors the groove width (Fig. 4b, d). Likely the electrostatic repulsion between M1-M3 and M4-M6 on the opposite sides of the groove contributes to the groove widening upon $Mg^{2+}$ saturation.

We can even further speculate that, in the apo form, inner-shell $Mg^{2+}$ ions M1–M6 may hold the incoming ligand at the groove by electrostatic attraction with its carboxy moiety. The 5′-deoxyadenosyl group would then explore the groove and get into the binding pocket via the pre-widened entryway.

While inner-shell $Mg^{2+}$ ions widen the ligand-entry groove for both of the liganded forms, the widening is greater for SAM. This greater widening can be attributed to the fact that SAM samples two conformations (Supplementary Fig. 3b, c). The groove is widened more when SAM adopts the L shape, where the aminocarboxypropyl group extends out from the groove into the solvent.

**Inner-shell $Mg^{2+}$ ions form outer-shell coordination with ligands and stabilize U16-ligand hydrogen bonding.** The aminocarboxypropyl group of the ligands is exposed to the ligand-entry groove (Fig. 2b). It is mobile in the MD simulations for SAH and extremely so for SAM due to its sampling of two different conformations (Supplementary Fig. 3b, c). The carboxy moiety can potentially form outer-shell coordination with three of the groove-lining $Mg^{2+}$ ions: M2, M5, and M6 (Figs. 4c and 5a). We calculated the shorter of the distances from the two oxygen atoms on the ligand carboxyl to each of these three $Mg^{2+}$ ions (Supplementary Fig. 10a, b). There are indeed substantial fractions of MD frames where the ligand carboxyl is within the 2.5 to 5 Å range for outer-shell coordination with M6, M5, and M2 (Fig. 5b). For the SAH-bound form, the outer-shell coordination fractions with these $Mg^{2+}$ ions are 77.7%, 34.7%, and 13.6%, respectively. Coordination with M6 and M5 can occur

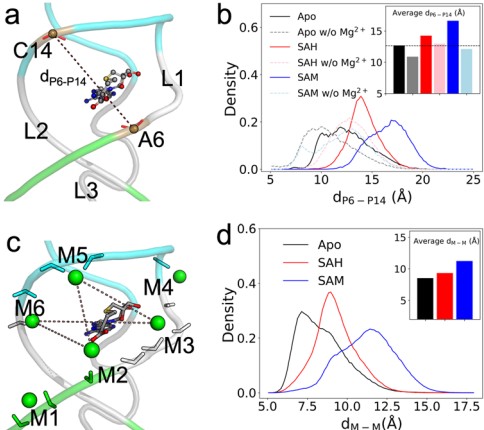

**Fig. 4 Groove widening upon ligand binding and by inner-shell $Mg^{2+}$ ions. a** The distance, $d_{P6-P14}$, for measuring the groove width. **b** Distributions of $d_{P6-P14}$ in the simulations of the three forms of the riboswitch without (labeled as "w/o") or with saturating $Mg^{2+}$. Inset: average values for six systems. **c** Six inner-shell $Mg^{2+}$ ions lining the two sides of the groove. **d** Distributions of the minimum distance between M2 and M3 on one side of the groove and M5 and M6 on the opposite side. Inset: average values for the three systems.

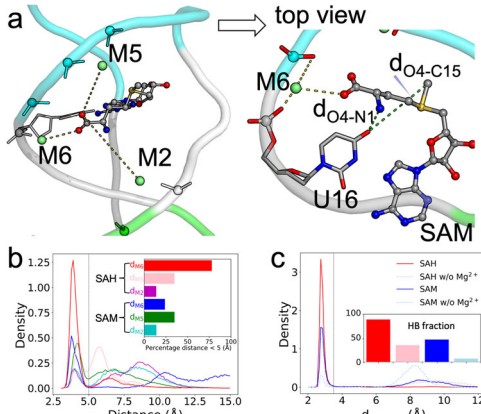

**Fig. 5 Outer-shell coordination with ligands and stabilization of U16-ligand hydrogen bond by M2, M5, and M6. a** The positions of M2, M5, and M6 relative to the ligand carboxy moiety. Top view: M6 forms inner-shell coordination with C15 and outer-shell coordination with both U16 and SAM carboxy. Also shown are the distances from the U16 O4 atom to the SAM N1 and C15 atoms. **b** Distributions of the distances between the ligand carboxyl and M2, M5, and M6. A vertical line at 5 Å indicates the cutoff for outer-shell coordination. Inset: fractions of frames forming outer-shell coordination. **c** Distributions of $d_{O4-N1}$. A vertical line at 3.5 Å indicates the cutoff for hydrogen bond formation. Inset: fractions of frames forming a U16-ligand N1 hydrogen bond.

simultaneously, but coordination with either M6 or M5 and coordination with M2 are mutually exclusive, as they are located on opposite sides of the groove. The high propensity that the ligand carboxy moiety forms outer-shell coordination with at least one of the groove-lining Mg²⁺ ions buttresses the foregoing speculation about their potential role in holding the ligand at the groove prior to binding.

For the SAM-bound form, the coordination fraction with M6 is significantly reduced, to 23.7%, while those with M5 and M2 are similar to the SAH counterparts. Whereas the SAH-bound form has similarly high propensities for outer-shell coordination with M6 in all the four replicate simulations, the SAM-bound form does so in only one (MD4) of the four replicate simulations (Supplementary Fig. 10a, b). Therefore the inner-shell Mg²⁺ ions stabilize ligand binding by forming outer-shell coordination with the ligand carboxyl. However, the stabilizing effect, specifically from M6, is greater for SAH than for SAM, thereby countering the general favorability of SAM (Fig. 2a) and diminishing potential selectivity against SAH.

We have noted above that U16 and G17 also make a greater contribution to the binding energy for SAH than to the counterpart for SAM (Supplementary Fig. 2a). The counter-actions of M6 revealed a moment ago and of U16 and G17 are closely linked, as we now explain. In the static NMR structure 6HAG, the O4 atom of U16 and the O6 atom of G17 are both near but outside the hydrogen-bonding distance (3.5 Å) from the amino N1 atom of the ligand[20]. However, in the MD simulations, $d_{O4-N1}$ (Fig. 5a, top view) frequently comes into the hydrogen-bonding range (Supplementary Fig. 10c, d). The fractions of MD frames forming the U16-ligand amino hydrogen bond are 88.8% for SAH and 48.1% for SAM (Fig. 5c). The higher hydrogen-bonding fraction for SAH accounts for the greater contribution of U16 to the binding energy for this ligand. Similarly, $d_{O6-N1}$ comes within 3.5 Å frequently (85.1%) for SAH but much less so (49.9%) for SAM, and hence a greater contribution of G17 to the binding energy of SAH. The O6-N1 contact does not always satisfy the angular requirement of a hydrogen bond (donor-H-acceptor angle > 120°), but the short distance still results in a significant van der Waals interaction.

For SAM, the methyl (C15 atom) on the sulfur center can potentially clash with the U16 O4 (Fig. 5a, top view) or G17 O6 atom. SAM switches between two conformations (Supplementary Fig. 3b, c), in which $d_{O4-N1}$ and $d_{O4-C15}$ are anticorrelated (Supplementary Fig. 10e; $r = -0.51$): in the U-shape conformation, N1 gets close to U16 O4 but C15 moves away, whereas in the L-shape conformation the opposite occurs. Therefore the methyl can interfere with U16-ligand amino hydrogen bonding, thereby explaining the much higher propensity of forming this hydrogen bond by SAH and the greater contribution of U16 to the binding energy of this ligand.

When this hydrogen bond with U16 is formed, it places the carboxy moiety in a position to form outer-shell coordination with M6. Therefore, when the methyl of SAM interferes with U16-ligand amide hydrogen bonding, it simultaneously interferes with M6-ligand carboxyl outer-shell coordination. In simulation MD4, where SAM forms M6-ligand carboxyl coordination with 94.9% probability (Supplementary Fig. 10b), it also forms the U16-ligand amide hydrogen bond with 98.6% probability (Supplementary Fig. 10d). In contrast, in simulations MD1–MD3 where coordination with M6 is never formed, the probability for the hydrogen bond drops to 31.3%. Moreover, while the M6-ligand carboxyl distance in these three simulations never reaches the outer-shell coordination cutoff, it shows a strong correlation with the U16-ligand amide distance (Supplementary Fig. 10b, d; $r = 0.87$).

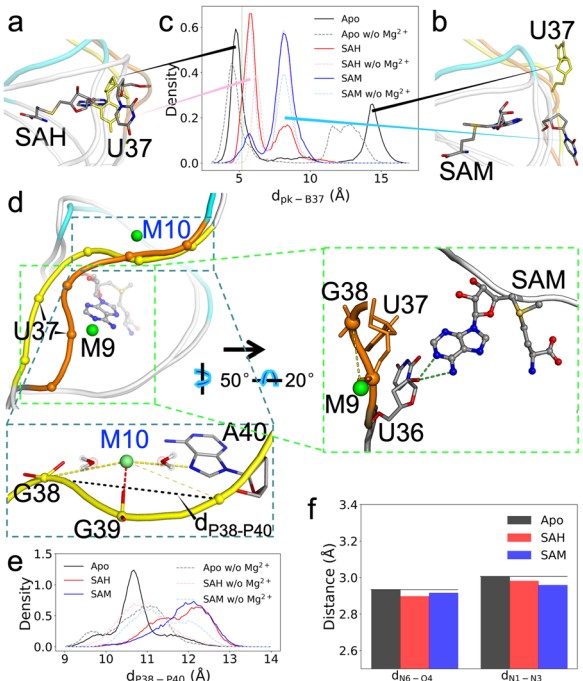

**Fig. 6 M10 facilitates the release of the SD sequence in the apo form.** **a** The U37 nucleobase is extruded upon ligand binding (C, N, and O atoms in gray, blue, and red, respectively), but can move into the ligand-binding pocket in the apo form (all yellow). **b** An alternative pose for U37, just outside the binding pocket in the liganded form but farther out in the apo form. **c** Distributions of $d_{pk-B37}$, the distance from the U37 nucleobase to the center of the binding pocket. **d** Backbone shape of nucleotides 36–40, in orange for the SAM-bound form and yellow for the apo form. Zoomed view of the SAM-bound form on the right: outer-shell coordination of M9 with G38 and hydrogen bonding between U36 2′-OH and SAM N4 or N5. Zoomed view of the apo form at the bottom: M10 forms inner-shell coordination with G39 and outer-shell coordination with both G38 and A40. Also indicated is the distance between the phosphates of G38 and A40. **e** Distributions of $d_{P38-P40}$. **f** Average distances between hydrogen-bonding donors and acceptors on the base-pair partners A40 and U12.

In short, the interactions of the ligand aminocarboxyl with Mg²⁺ ion M6 and the U16 and G17 nucleobases are highly cooperative, with a strong tendency to form or break at the same time. As a last testament to this cooperativity, without Mg²⁺, the probabilities for forming the U16-ligand amino hydrogen bond are dramatically reduced for both SAH and SAM (Fig. 5c and Supplementary Fig. 10d).

**An inner-shell Mg²⁺ ion in the apo form facilitates the release of the SD sequence**. Upon ligand binding, the U37 nucleobase is extruded from the ligand-binding pocket. In the MD simulations, this base adopts two alternative poses (Fig. 6a, b). We monitored the movement of this nucleobase by calculating its distance ($d_{pk-B37}$) from the center of the binding pocket (defined by the pocket-lining bases C8, G9, U16, G17, and G38) (Fig. 6c). The distribution of $d_{pk-B37}$ exhibits two peaks, corresponding to the two alternative poses, for either the SAH or SAM-bound form. These alternative poses explain why its RMSF is higher than those of its immediate neighbors (Supplementary Fig. 4b). The upstream neighbor, U36, tends to form a hydrogen bond, via its 2′-OH, with either the N4 or N5 atom of the ligand base (Fig. 6d, zoomed view on the right). The downstream neighbor, G38, base pairs with C15 as part of stem S2.

In the simulations of the apo form, the U37 nucleobase moves either into the binding pocket (Fig. 6a; $d_{pk-B37} \sim 4.7$ Å) or far away from the binding pocket (Fig. 6b; $d_{pk-B37} \sim 14.4$ Å), spending roughly equal times in the two positions (Fig. 6c). The large distance between these two positions of U37 contributes to the conformational heterogeneity of the apo form.

The ligand-forced extrusion of U37 and the stacking of G38 against the ligand base (Fig. 1c) result in changes in both the backbone curvature and the $Mg^{2+}$ coordination pattern on the downstream side. Relative to the apo form, the phosphates of U37 and G38 are brought closer, and the backbone of nucleotides G38-G39-A40 is straightened (Fig. 6d and zoomed view at the bottom). We measured the latter effect by calculating the distance, $d_{P38-P40}$, between the phosphorus atoms of G38 and A40 (Fig. 6e). The peak distance increases from 10.6 Å in the apo form to 12.4 Å in both of the liganded forms.

$Mg^{2+}$ ion M9 forms inner-shell coordination with the U37 phosphate in all the three forms of the riboswitch (Supplementary Fig. 9 and Fig. 6d). M9 also forms outer-shell coordination with G38 part of the time in the liganded forms due to the closer U37-G38 distance (Supplementary Fig. 9, and Figs. 3b, 6d, zoomed view on the right), but not all in the apo form. Instead, the curved backbone of apo G38-G39-A40 creates an inner-shell $Mg^{2+}$ site that is unique to the apo form. M10 forms inner-shell coordination with G39 and outer-shell coordination with both G38 and A40; for A40, outer-shell coordination can occur via either the phosphate or the base N7 atom (Supplementary Fig. 9 and Fig. 6d, zoomed view at the bottom). Note that the apo G38-G39-A40 backbone is curved and M10 is bound all the time, regardless of whether U37 takes its position inside or away from the binding pocket; the distribution of $d_{P38-P40}$ has only a single peak. Therefore the curved shape may be intrinsic to G38-G39-A40, stabilized by the inner-shell $Mg^{2+}$ ion M10. Without $Mg^{2+}$, the $d_{P38-P40}$ distribution in the apo form no longer shows distinction from those in the liganded forms (Fig. 6e).

Because the M10 site exists exclusively in the apo form and resides completely inside the SD sequence, we suspected that it might play a direct functional role. One possibility is that M10 maintains G38-G39-A40 in an autonomous mode such that their base-pairing with the upstream nucleotides C15-C14-U12 is weakened. Figure 6f shows that, indeed, for the base pair between A40 and U12, the donor-acceptor distances are longer in the apo form. Therefore M10 can directly facilitate the release of the DS sequence to initiate translation.

## Discussion

We have carried out extensive MD simulations to investigate the essential roles of $Mg^{2+}$ in the ligand binding and conformational transition of the SAM/SAH riboswitch. We found 11 inner-shell $Mg^{2+}$ ions each in the apo form and the SAM and SAH-bound forms. Six of the common $Mg^{2+}$ ions (M1 to M6) line the ligand-entry groove to widen it, thereby pre-organizing the riboswitch for ligand binding. M2, M5, and M6 alternately form outer-shell coordination with the ligands. In addition, M6 stabilizes U16 and G17-ligand amide interactions. These interactions occur with reduced probability for SAM due to the interference of its methyl, thereby countering the general favorability of this ligand over SAH and diminishing the selectivity between these two ligands. One $Mg^{2+}$ ion, M10, unique to the apo form maintains the SD sequence in a curved conformation and weakens its base-pairing with upstream nucleotides, thereby facilitating its release for ribosome binding.

Key aspects of our MD results are validated by experimental observations. For example, the flexibility profiles calculated from the MD simulations agree well with $^1H-^{13}C$ hetNOE data[20]. The mobility of the ligand carboxy moiety seen in our MD simulations is supported by its different orientations in the NMR structure 6HAG and crystal structure 6YL5[20,21] (Supplementary Fig. 11a). The probabilities of different RNA atoms forming inner and outer-shell coordination with $Mg^{2+}$ match those tabulated from crystal structures[39]. As further validation, we compared the three inner-shell $Na^+$ ions in 6YL5 with our inner-shell $Mg^{2+}$ sites. One $Na^+$ ion forms inner-shell coordination with the ligand carboxyl and outer-shell coordination with A6 and A7 (env9b riboswitch numbering); this $Na^+$ is similar to our M2. The second $Na^+$ ion forms inner-shell coordination with G20 and outer-shell coordination with C19, close to our M8. The third $Na^+$ ion forms inner-shell coordination with G39 and outer-shell coordination with A40 (via both phosphate and N7). Interestingly, this $Na^+$ ion is very much like our M10, except that ours is found in the apo form. Comparing backbone shapes of the nearby nucleotides, we find that 6LY5 is more curved than 6HAG (Supplementary Fig. 11a), to a similar extent as our apo form (Supplementary Fig. 11b). It looks as if the removal of the flexible L3 in the crystal structure of the SAH-bound form reduces restraints on G39 and A40 so they behave as if in the apo form.

Our MD simulations have generated unique mechanistic insights. In particular, the solvent exposure of the methyl on SAM leads to similar interaction energies for SAM and SAH with the riboswitch nucleotides, yet the positive charge on the sulfur center still generally favors SAM over SAH. It is $Mg^{2+}$ ions that provide compensatory effects for SAH and dimmish the selectivity between the two ligands. Without the methyl, M6 is better able to form outer-shell coordination with the SAH carboxyl and stabilize the U16 and G17-SAH amide interactions. We also show that $Mg^{2+}$ ions widen the ligand-entry groove so to reduce the energy barrier for entering the ligand-binding pocket. We further speculate that these $Mg^{2+}$ ions can potentially hold the ligand, via outer-shell coordination with its carboxy moiety, to give the ligand more chance to explore the groove and enter the ligand-binding pocket. Lastly, while $Mg^{2+}$ ions are generally known to stabilize RNA structures including helical elements, our characterization of M10 in the apo form shows that they can also specifically interact with one strand of a helical element and peel it away from the complementary strand. The end result, for the SAM/SAH riboswitch, is the release of the SD sequence.

Despite their essential roles illustrated here, $Mg^{2+}$ is difficult to identify and can thus be dubbed the "dark" metal ion in RNA research. The present study, along with our previous work[48], has demonstrated that placing $Mg^{2+}$ ions initially according to a structure-based prediction method such as MCTBI[42] is an effective protocol for producing inner-shell ions in conventional MD simulations. We hope that this protocol and further developments will make MD simulations an even more powerful technique for characterizing both the structural determinants and the functional consequences of $Mg^{2+}$ coordination.

## Computational methods

**Preparation of RNA systems.** The initial SAH-bound structure of the SAM/SAH riboswitch was from the NMR structure 6HAG[20]. The SAH ligand was replaced by SAM to generate the SAM-bound structure and removed to generate the apo structure. The original hydrogen atoms on the RNA molecule were removed and re-added by using the Leap module in AMBER18[52]. For each of the three forms (SAH or SAM-bound or apo) of the riboswitch, four types of $Mg^{2+}$ initial placement were applied (Supplementary Table 1): (i) without any $Mg^{2+}$; (ii) addition of 25 $Mg^{2+}$ ions at sites predicted by MCTBI[42]; (iii) addition of 21 $Mg^{2+}$ ions (enough to neutralize the RNA molecule) around the RNA as part of the solvation step using Leap; and (iv) addition of

41 $Mg^{2+}$ ions in the solvent using Leap. Type (iv) was applied to all the 10 models in 6HAG for the two liganded forms; in all other cases only model 5 was prepared. In the solvation step, the RNA [plus $Mg^{2+}$ in the case of type (ii)] was placed in a truncated octahedron periodic box of TIP3P[54] water molecules. The number of water molecules was approximately 15580. Neutralizing $Na^+$ or $Cl^-$ ions and 0.15 M NaCl [except type (i)] were also added. In (ii) and (iii), $Mg^{2+}$ was added around the RNA whereas $Na^+$ was placed into the solvent; in (iv), the opposite was true: $Na^+$ was added around the RNA whereas $Mg^{2+}$ was placed into the solvent. The distance from the RNA molecule to the edge of the box was at least 12 Å.

The force field for RNA was an improved version of AMBER ff99[55], with correction for α/γ dihedrals (bsc0)[56] and correction for χ dihedrals ($\chi_{OL3}$)[57]. The parameters for $Mg^{2+}$ were from Li et al.[58]; those for $Na^+$ and $Cl^-$ were from Joung and Cheatham[59]. To generate force-field parameters for the ligands, the structures of the ligands were optimized using the Gaussian16 program[60] at the HF/6-31 G* level. The atomic partial charges were assigned using the restrained electrostatic potential (RESP) method[61]; other parameters were taken from the general Amber force field[62].

**Molecular dynamics simulations**. Energy minimization and MD simulations were carried out using the AMBER18 package[52]. To start, each system was minimized by the steepest-descent and conjugate-gradient methods, each for 2500 steps. The preparatory stage of the simulation consisted of 50 ps of temperature ramping from 100 K to 300 K at constant volume, 50 ps at 300 K and constant volume, and 50 ps at 300 K and 1.0 atm pressure while restraining the RNA and ligand atoms with a force constants of 5 kcal/(mol·Å²). The equilibration stage was 1 ns at constant temperature and pressure (without restraints). The production run was carried out in four replicates for 1 μs at constant temperature and pressure. The temperature (300 K) was regulated by the Langevin thermostat[63], and pressure (1.0 atm) was regulated by the Berendsen barostat[64]. Long-range electrostatic interactions were treated by the particle mesh Ewald method[65] with a direct-space cutoff of 12 Å. Bonds involving hydrogen atoms were constrained by the SHAKE algorithm[66]. The time step was 2 fs. Frames were saved at 100 ps intervals for later analysis.

**Interaction energies between individual nucleotides and the ligand**. The interaction energies were calculated by the MM/GBSA method[53] using AMBER18. Results were obtained for 5000 frames in the second 500 ns of each replicate simulation, and then averaged over four replicate simulations (or further over 10 starting models).

**Other analyses**. All other analyses were done on 10000 frames in the entire 1000 ns of each replicate simulation and then averaged over four replicate simulations. RMSFs were calculated by first aligning using backbone atoms (P, O3′, O5′, C3′, C4′, C5′, excluding L3) of the riboswitch to obtain an average structure and then finding the deviations of a specific set of atoms from the average structure. The set of atoms was: (i) H1′ and C1′; or (ii) H6 and C6 (C and U nucleotides) or H8 and C8 (A and G nucleotides). RDFs were calculated from the number of $Mg^{2+}$ ions within a distance bin (0.05-Å width) from a given RNA atom, normalized by the expected number of $Mg^{2+}$ ions in that bin assuming uniform density. RMSFs, RDFs, distances, and hydrogen bond formation were calculated by the CPPTRAJ program[67]. Hydrogen bonding criteria were: donor-acceptor distance <3.5 Å and donor-H-acceptor angle >120°.

The fraction of frames where a nucleotide made either inner or outer-shell coordination with $Mg^{2+}$ ions was calculated using a Tcl script in VMD[68], based on distances between the phosphate OP1 or OP2 atom and any $Mg^{2+}$ ion, with cutoffs at 2.5 and 5 Å, respectively. The 2.5-Å cutoff was chosen because it falls well into the gap between the first and second peaks of the RDFs around OP1 and OP2 (Supplementary Fig. 3); The 5-Å cutoff was chosen because it is where the second peak of the RDFs falls to a minimum. The densities of $Mg^{2+}$ were determined using a python script importing the MDAnalysis package[69].

**Statistics and reproducibility**. Molecular dynamics simulations were carried out in four replicates for each system, for one NMR model as starting structure in most cases but for 10 NMR models in one case (Supplementary Table 1). The mean and standard deviation were calculated using the replicate simulations in the former cases but among results from the 10 starting models in the latter case. Convergence of the MD simulations was validated by comparing the results calculated in 250-ns blocks along the trajectories.

**Reporting summary**. Further information on research design is available in the Nature Portfolio Reporting Summary linked to this article.

## Data availability

All data generated or analyzed during this study are included in this published article (and its supplementary data files). The source data for all the plots presented in figures are deposited in GitHub at https://github.com/hzhou43/SAM_SAH-riboswitch.

## Code availability

Data analysis procedures were described under Computational Methods. All the computer programs used were cited and publicly available. The input files for MD simulations of the SAM/SAH riboswitch in the apo and two liganded forms and the initial and final coordinate files are deposited in GitHub at https://github.com/hzhou43/SAM_SAH-riboswitch.

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

## Acknowledgements

This work was partially supported by funding from the Natural Science Foundation of Shandong Province (ZR2019MA040 to G.H.), the National Natural Science Foundation

of China (32171249 and 62071085 to G.H.), and the US National Institutes of Health (GM118091 to H.X.Z.).

## Author contributions

G.H. and H.X.Z. designed research, conducted research, analyzed data, and wrote manuscript.

## Competing interests

The authors declare no competing interests.

## Additional information

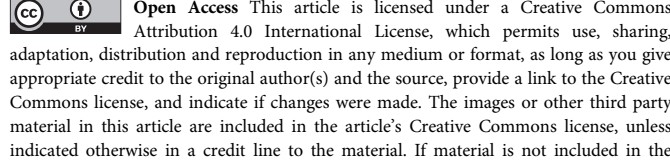

