## [Peer Review File · Communications Biology]

Reviewers' comments:

Reviewer #1 (Remarks to the Author):

The author conducted large-scale MD simulations of SAM/SAH riboswitch using mature force fields and discussed the influence of magnesium ions on ligand binding. Some novel viewpoints were proposed, such as the repulsive effect of inner-shell magnesium ions at the entrance of ligand pockets promoting ligand entry. This manuscript has some innovation and the results are also reliable. However, I still have some issues that the author needs to address, which are listed below:

1. Why is there almost no inner-shell magnesium ions in the Leap(41) system, while there is a significant increase in the Leap(21) system? How to determine if it is caused by ion competition? Because the ion allocation method of the LEAP module has a high probability of allocating the initial position of ions very close to the RNA backbone, resulting in insufficient hydration opportunities for magnesium ions, it is necessary to rule out the impact of these two systems caused by the initial ion allocation scheme.
2. If magnesium ions undergo an outer-shell to inner-shell transition in the Leap(21) system, then this will be a very interesting phenomenon. It should be meaningful to demonstrate the details of magnesium ion dehydration in this system.
3. The manuscript states that metal ions mainly gather around the first 21 residues, is this because these regions have lower electrostatic potential? Can the authors provide some characterizations of electrostatic potential surfaces?
4. Do all calculations of binding free energy come from the Leap(41) system? Can the authors provide the binding free energy results in the MCTBI and Leap(21) systems? I would like to know if the molecules that bind in the inner-shell mode near the binding pocket have an impact on the ligand binding free energy, as described in the "Inner-shell Mg²⁺ ions form outer-shell coordination with ligands and stabilize U16-ligand hydrogen bonding" viewpoint later (page 10, line 259).
5. The simulation results of outer-shell magnesium ions (4.3 Å peak in RDF) are in good agreement with previous literature (Physical Review E, 99:012420, 2019). However, is it possible that the viewpoint of "Inner-shell coordination is nearly exclusively formed with OP1 and OP2, with OP2 favored about 2-fold over OP1" (Page 8, line 200) comes from statistical errors? Because this situation does not occur in the RDF of the SAM system (Figure S3).
6. In Figure 4B, it can be observed that as long as magnesium ions bind to the ligand entrance, it will to some extent cause the entrance to widen. However, the chemical structure and size of SAM and SAH are very similar. Why is the width at the entrance of SAM significantly larger than that of SAH when they are bound? Is it because of the difference in binding mode?
7. I noticed that the author conducted multiple repeated simulations for each case. are all the results (such as RMSF, groove width $\langle d \rangle_{p6-p14}$) in the manuscript based on the statistical average of these cases?

Additionally, there are some minor issues:

1. Note that there are some misleading writing styles in the text, such as 0A, 0B, 0C, etc. (Page 3). What do they represent?
2. "Via these direct and indirect interactions" can be changed into "Through these direct and indirect interactions". (Page 4, line 66)

Reviewer #2 (Remarks to the Author):

The authors have performed extensive MD simulations of SAM/SAH riboswitch to understand the role of Mg²⁺ ions in the ligand interactions and overall dynamics. The study relies on the Mg²⁺ binding sites predicted by a tool MCTBI and build the MD simulations based on that. The results from MD simulations have been analyzed accordingly by the authors to come to the conclusions. Although, the issues associated with long exchange rates in running MD simulations of RNA with Mg²⁺ ions has been assumed to be insignificant. Given that, authors have performed extensive sets of MD to get the best information possible. The reviewer has both major and minor comments for the authors.

Major comments to authors:

- The authors should refrain from stating that MD simulations "identified" inner-shell Mg²⁺ ions as it is not 100% correct.
- The enhanced sampling approaches by two groups (Thirumalai and Mackerell) have been recently shown to address the issues for long exchange rates to identify the Mg²⁺ binding sites. Please discuss it accordingly.
- Please comment on the role of Na⁺ ions in and around the binding sites.
- Does the ionic environment follow counterion condensation theory to neutralize the RNA within certain distance?
- Authors should comment on the results from the systems where Mg²⁺ ions were added with Leap protocol. What were major differences?

Minor comments:

P3 L44 - The terminology "A high resolution NMR structure" should be reconsidered.

P3 L46 - Please refer to figure 1 here.

P8 L191 - Do authors mean to state "not a single new inner-shell Mg²⁺ ion wa found"?

P8 L200 - "OP2 favored about 2-fold over OP1" How is this significant? How are the two oxygens differentiated? What kind of environment causes this?

P10 L254-257 Do authors have any references that support such a speculation?

P15 L397 - In one of the papers by MacKerell lab, it was shown that Mg²⁺ impacts RNA folding by push-pull mechanism. Can authors discuss on that based on this?

P16 L427 - Monovalent ions support the divalent ions in various conditions. Usually 0.15 M concentration is used for NaCl/KCl. how was this concentration translated into number of ions? based on volume of the box? Was the volume corrected for the volume occupied by the RNA? what would be effective concentration in the box?

P18 L475 - Please explain what is baseline?

We thank the reviewers for their constructive comments. Our point-by-point response is given below in blue.

Reviewer #1 (Remarks to the Author):

The author conducted large-scale MD simulations of SAM/SAH riboswitch using mature force fields and discussed the influence of magnesium ions on ligand binding. Some novel viewpoints were proposed, such as the repulsive effect of inner-shell magnesium ions at the entrance of ligand pockets promoting ligand entry. This manuscript has some innovation and the results are also reliable. However, I still have some issues that the author needs to address, which are listed below:

1. Why is there almost no inner-shell magnesium ions in the Leap(41) system, while there is a significant increase in the Leap(21) system? How to determine if it is caused by ion competition? Because the ion allocation method of the LEAP module has a high probability of allocating the initial position of ions very close to the RNA backbone, resulting in insufficient hydration opportunities for magnesium ions, it is necessary to rule out the impact of these two systems caused by the initial ion allocation scheme.

Indeed, the difference between Leap(21) and Leap(41) is largely due to the initial ion allocation scheme. Specifically, in Leap(21), Mg²⁺ ions were given the preference to be placed close to the RNA while Na⁺ ions were relegated to the solvent; in Leap(41) the preferences were reversed between Mg²⁺ and Na⁺. We now further clarify this difference between Leap(21) and Leap(41) (p. 6, 9, and 18).

2. If magnesium ions undergo an outer-shell to inner-shell transition in the Leap(21) system, then this will be a very interesting phenomenon. It should be meaningful to demonstrate the details of magnesium ion dehydration in this system.

All the inner-shell Mg²⁺ ions in Leap(21) were indeed in outer-shells or farther away from RNA phosphates. The transitions occurred very early on, during the energy minimization and heating stage of the simulation. In the new supplementary figure S5 and new movie S1, we demonstrate the transition into inner-shell coordination of a Mg²⁺ ion, initially at an initial distance of 6.9 Å from the nearest OP atom.

3. The manuscript states that metal ions mainly gather around the first 21 residues, is this because these regions have lower electrostatic potential? Can the authors provide some characterizations of electrostatic potential surfaces?

Yes, these regions have the most negative electrostatic potential (p. 10). We now present the electrostatic potential surface of the RNA in new supplementary figure S7.

Do all calculations of binding free energy come from the Leap(41) system? Can the authors provide the binding free energy results in the MCTBI and Leap(21) systems? I

would like to know if the molecules that bind in the inner-shell mode near the binding pocket have an impact on the ligand binding free energy, as described in the " Inner-shell Mg^{2+} ions form outer-shell coordination with ligands and stabilize U16-ligand hydrogen bonding" viewpoint later (page 10, line 259).

The binding free energy results shown in Fig. 2A were from the Leap(41) protocol. We now present the corresponding results for the MCTBI and Leap(21) protocols in new supplementary figure S2 (p. 7), and discuss these results in reference to U16-ligand hydrogen bonding (p. 12-14).

5. The simulation results of outer-shell magnesium ions (4.3 Å peak in RDF) are in good agreement with previous literature (Physical Review E, 99:012420, 2019). However, is it possible that the viewpoint of "Inner-shell coordination is nearly exclusively formed with OP1 and OP2, with OP2 favored about 2-fold over OP1" (Page 8, line 200) comes from statistical errors? Because this situation does not occur in the RDF of the SAM system (Figure S3).

We agree that, in the SAM form, OP2 is not as strongly favored over OP1 as in the apo and SAH form, and have revised the wording to: "with OP2 favored over OP1 by 1.3 to 3.0-fold (Figure S6)."

6. In Figure 4B, it can be observed that as long as magnesium ions bind to the ligand entrance, it will to some extent cause the entrance to widen. However, the chemical structure and size of SAM and SAH are very similar. Why is the width at the entrance of SAM significantly larger than that of SAH when they are bound? Is it because of the difference in binding mode?

Yes, the larger groove widening in the SAM form is due to its different binding characteristics from SAH. In essence, SAH has a single conformation, whereas SAM has two alternative conformations (new supplementary figure S3), one is similar to that of SAH and in the other the aminocarboxypropyl group extends into the groove. In the latter case the groove is wider (p. 11).

7. I noticed that the author conducted multiple repeated simulations for each case. are all the results (such as RMSF, groove width d_{p6-p14}) in the manuscript based on the statistical average of these cases?

The reported results for each system were based on the average of the multiple repeated simulations for that system.

Additionally, there are some minor issues:

1. Note that there are some misleading writing styles in the text, such as 0A, 0B, 0C, etc. (Page 3). What do they represent?

These are typos, which we have corrected them to be Figure 1A, 1B, 1C.

2. “Via these direct and indirect interactions” can be changed into “Through these direct and indirect interactions”. (Page 4, line 66)

We have now made the suggested change.

Reviewer #2 (Remarks to the Author):

The authors have performed extensive MD simulations of SAM/SAH riboswitch to understand the role of Mg²⁺ ions in the ligand interactions and overall dynamics. The study relies on the Mg²⁺ binding sites predicted by a tool MCTBI and build the MD simulations based on that. The results from MD simulations have been analyzed accordingly by the authors to come to the conclusions. Although, the issues associated with long exchange rates in running MD simulations of RNA with Mg²⁺ ions has been assumed to be insignificant. Given that, authors have performed extensive sets of MD to get the best information possible. The reviewer has both major and minor comments for the authors.

Major comments to authors:

- The authors should refrain from stating that MD simulations “identified” inner-shell Mg²⁺ ions as it is not 100% correct.

We have changed “identified” to “predicted” or “found”.

- The enhanced sampling approaches by two groups (Thirumalai and Mackerell) have been recently shown to address the issues for long exchange rates to identify the Mg²⁺ binding sites. Please discuss it accordingly.

We now cite these studies (new refs. 27 and 28; p. 4 and 5).

- Please comment on the role of Na⁺ ions in and around the binding sites.

Na⁺ ions do not form tight, site-specific coordination with RNA phosphates and are very mobile, even when the phosphates are not coordinated with Mg²⁺ [as in Leap(41); p. 9]. So Na⁺ ions mostly act as diffuse counterions.

- Does the ionic environment follow counterion condensation theory to neutralize the RNA within certain distance?

Qualitatively, the ion environment in the MCTBI protocol agrees with what is envisioned in the counterion condensation theory, in that over 10 Mg²⁺ ions tightly

coordinate with RNA phosphates and neutralize a large fraction of the RNA charge. The situation in the Leap(41) protocol is different. Here Na⁺ ions are the species nearest to RNA phosphates but are very mobile, such that there is no clear demarcation between a “condensation” zone and a diffuse zone.

- Authors should comment on the results from the systems where Mg²⁺ ions were added with Leap protocol. What were major differences?

Again, the difference between Leap(21) and Leap(41) is largely due to the initial ion allocation scheme. Specifically, in Leap(21), Mg²⁺ ions were given the preference to be placed close to the RNA while Na⁺ ions were relegated to the solvent; in Leap(41) the preferences were reversed between Mg²⁺ and Na⁺. We now further clarify this difference between Leap(21) and Leap(41) (p. 6, 9, and 18).

Minor comments:

P3 L44 - The terminology “A high resolution NMR structure” should be reconsidered.

We now remove “high resolution”.

P3 L46 - Please refer to figure 1 here.

We have now corrected the typo “0A” that should have been Figure 1A.

P8 L191 - Do authors mean to state “not a single new inner-shell Mg²⁺ ion was found”?

Yes, due to the fact that Na⁺ ions “pre-occupied” the phosphate sites, as we now further clarify (p. 9).

P8 L200 - “OP2 favored about 2-fold over OP1” How is this significant? How are the two oxygens differentiated? What kind of environment causes this?

We now explain that OP2 typically points toward the nucleic acid base whereas OP1 toward the solvent (p. 9). This difference explains the preference of OP2 over OP1 for inner-shell coordination. This preference is also seen in the statistics collected by Zheng et al. from crystal structures.

P10 L254-257 Do authors have any references that support such a speculation?

It is only our speculation, but supported by our observation of frequent carboxyl-Mg²⁺ coordination in the bound form (p. 12).

P15 L397 - In one of the papers by MacKerell lab, it was shown that Mg²⁺ impacts RNA folding by push-pull mechanism. Can authors discuss on that based on this?

We now cite the Kognole and MacKerell paper (ref. 28), in the contexts of RNA folding (p. 4) and enhanced sampling (p. 5). The context noted by the reviewer, i.e., compensatory effects between SAM and SAH, however, has no direct relation with the push-pull mechanism.

P16 L427 - Monovalent ions support the divalent ions in various conditions. Usually 0.15 M concentration is used for NaCl/KCl. how was this concentration translated into number of ions? based on volume of the box? Was the volume corrected for the volume occupied by the RNA? what would be effective concentration in the box?

We calculated the number of ions for a given salt concentration according to the number of water molecules: $N_{\text{ion}} = 0.0187 \times \text{conc (M)} \times N_{\text{water}}$.

P18 L475 - Please explain what is baseline?

We have modified this sentence to state that 5 Å is where the second peak of the RDFs falls to a minimum.

REVIEWERS' COMMENTS:

Reviewer #1 (Remarks to the Author):

The author has fully answered all my questions, therefore I agree to publish this manuscript

Reviewer #2 (Remarks to the Author):

Authors have satisfactorily addressed reviewer's comments.